# Synthesis of Diabetic Retina Fundus Images
# Using Semantic Label Generation

**Joon-Ho Son**                                                                    JOON-HO.SON17@IMPERIAL.AC.UK
**Amir Alansary**                                                                  A.ALANSARY14@IMPERIAL.AC.UK
**Daniel Rueckert**                                                                D.RUECKERT@IMPERIAL.AC.UK
**Bernhard Kainz**                                                                 B.KAINZ@IMPERIAL.AC.UK
**Benjamin Hou**                                                                   BENJAMIN.HOU11@IMPERIAL.AC.UK
*Department of Computing, Imperial College London, UK*

**Editors:** Under Review for MIDL 2021

## Abstract

Automatic segmentation of retina lesions have been a long standing and challenging task for learning based models, mostly due to the lack of available and accurate lesion segmentation datasets. In this paper, we propose a two-step process for generating photo-realistic fundus images conditioned on synthetic "ground truth" semantic labels, and demonstrate its potential for further downstream tasks, such as, but not limited to; automated grading of diabetic retinopathy, dataset balancing, creating image examples for trainee ophthalmologists, etc.

**Keywords:** Image Synthesis, Segmentation, Generative Adversarial Networks, Diabetic Retinopathy

## 1. Introduction

Retina fundus imaging is a modality that's quick and easy to obtain, however their usefulness is limited by the availability of human graders. With advancements in deep learning, data driven models may facilitate the process of diagnosis, alleviating this bottleneck. Despite this, training robust machine learning models requires manually annotated semantic labels en-masse, which are costly to obtain, and remains to be one of the primary obstacles in producing generalisable models. This challenge can be viewed in two aspects: the scarcity of manually annotated ground truths such as lesion segmentation maps, and class imbalance in available datasets. Both of these can be seen in datasets designed for classification and segmentation tasks related to diabetic retinopathy (DR).

Recently, the idea of leveraging generative models such as GANs to combat data scarcity by synthesising artificial retina images has gained attention, with DR-GAN (Zhou et al., 2020b) representing the current state-of-the-art by introducing a network that enables finer control over the generation of pathological cases. However, this model still relies on pre-existing structural and lesion semantic labels. Meanwhile, generation of natural images via texturing semantic images has been explored by Volokitin et al. 2020. Building on these ideas, this work explores a two-step process to generate photo-realistic retinas with accompanying pixel-wise semantic labels. First, synthetic semantic labels for retina structures and lesions are generated from random noise, conditioned on DR severity. These are then subsequently used to condition an image-to-image translation model to create the final retina fundus image.

## 2. Methods

We use a modified ACGAN architecture (Odena et al., 2017) to generate class-conditioned semantic labels at a resolution of $256 \times 256$, which were then upsampled to $512 \times 512$. The generator produces feature maps of shape $C \times H \times W$, where each channel corresponds to a specific lesion class. The sparse nature of the semantic images, coupled with the large number of channels, made the discriminator's job very easy, causing the GAN to be unstable and prone to collapse early in training. To mitigate this, two generator training iterations were run for each discriminator iteration, and label smoothing with adaptive discriminator augmentation (Karras et al., 2020) was applied to prevent overfitting. To synthesise the retinas from the generated semantic labels, we use the SPADE (Park et al., 2019) image-to-image translation model, generating images at a resolution of $512 \times 512$.

## 3. Experiments

 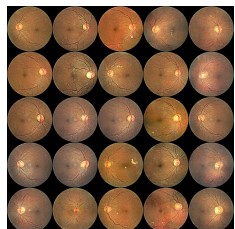

| Synthetic Data | | Precision | Recall | $F_1$ |
|---|---|---|---|---|
| Percentage | Count | | | |
| 0% | 0 | 0.5777 | 0.4692 | 0.4694 |
| 25% | 246 | 0.5860 | 0.4938 | 0.4965 |
| 50% | 738 | 0.5925 | 0.4727 | 0.4824 |
| 75% | 2214 | 0.6569 | 0.3990 | 0.4522 |

$(a)$ Semantic labels $\quad$ $(b)$ Fundus images $\quad$ $(c)$ # of synthetic images added with real data

Figure 1: (a/b) Synthetically generated labels and subsequent corresponding fundus images (columns L-to-R: DR Grade 0 to 4). (c) Segmentation performance of models with addition of synthetic data.

The models are trained and evaluated on a combination of IDRiD (Porwal et al., 2018) and FGADR (Zhou et al., 2020a) datasets. A number of images were omitted from FGADR, as they are inconsistent with the other samples due to annotator bias. To prevent data leakage from the generative models, a test set consisting of a random sample of 20% of the total data was held out. Since samples from the IDRiD dataset do not have accompanying DR grades, predicted grades on these images for weakly supervised learning are used.

Our preliminary experiments focus on generating semantic labels for only the optic disc and hard exudates. Layout prediction of the optic disc in particular was important for visually plausible retinas, where without, the SPADE generator created images which exhibited optic discs with poorly defined boundaries.

Figures 1(a) and 1(b) show samples of synthetically generated semantic labels and fundus images, respectively. The visual results demonstrate that by dividing the task into simpler sub-problems, the GANs are able to successfully learn relationship between DR grades and prevalence of hard exudate lesions. However, a number of failure cases are still exhibited, noticeably in the form of generating multiple optic discs. Moreover, when the optic disc is obscured, the image model is unable to correctly place the root of the vessels.

To evaluate the results quantitatively and measure the impact of synthetic data on downstream tasks, such as lesion segmentation, synthetic retina samples are uniformly generated across DR grades 0 through 4. Using a basic U-Net architecture (Ronneberger et al., 2015), a number of models were trained with varying amounts of synthetic data in

addition to real data. Each model was trained for 40 epochs (where performance plateaued). The final evaluation was performed on the held-out test dataset, with results shown in Table 1(c). From this, we can see that using small amounts of synthetic data yields the greatest gain in overall performance, as measured by the F-score. An interesting trend is exhibited where increasing the amount of synthetic data increases precision while decreasing recall, possibly caused by a bias in the generated data. The overall effect of this is that F-score increases until a critical point where gains in precision are offset by losses in recall.

## 4. Conclusion

In this work, we have shown that controlled synthesis of retina fundus images shows promising results for downstream medical imaging tasks. Further applications yet to be explored include improving the performance of DR severity prediction models, and use in training human experts to better identify retinal lesions.

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
