# OpenReview forum: "Synthesis of Diabetic Retina Fundus Images Using Semantic Label Generation"
_MIDL.io/2021/Conference/Short — MIDL 2021 Poster_

### Official Review · Reviewer_z39b · 2021-04-22

**Confidence:** 5
**Final Rating:** 4

**Summary:**

In this paper, the authors propose a two-step process for generating photo-realistic fundus images conditioned on synthetic “ground truth” semantic labels, and demonstrate its potential for further downstream tasks as listed by the authors.
The main problem involves lesion segmentation in fundus photographs, which can be challenging because of scarce data available for training DL algorithms, and annotators' bias in the GT. To augment training datasets, the authors propose a method which has two main contributions: 1) generate photo realistic fundus images conditioned on lesion GT; 2) demonstrate potential for downstream tasks.



**Strengths:**

Opposed to DRGAN which requires annotation of lesions; the method proposed here allows to generate plausible lesions and optic disk (OD) from noise, conditioned on disease severity. These are then used to condition the generation of fundus images. The quality of the generated images is assessed in a downstream segmentation task, where the synthetic images are introduced to the training set ad different rates.

**Weaknesses:**

In this first attempt, semantic labels are limited to OD and hexudates. Do you plan to produce macula, blood vessels as part of the semantic labels in the future? I think that especially the generation of blood vessels can be particularly challenging as the vascular-semantic label must obey some haemodynamic rules, e.g. proper arterial/venous distribution, retinal tissue perfusion, and embedding these constraints in a DL framework can be difficult but necessary.

The paper is well described, still some points are unclear: What is the discriminator discriminating? The type of lesion or real/fake image/ lesion? Is SPADE trained end to end with the other portion GAN that produces lesions? Or is SPADE trained and used at a second stage, after the GAN for lesions has been trained?

The code is not provided.


**Deanonymize Review:**

no

**Detailed Comments:**

I think it’s a good step the fact that when multiple ODs were placed, the model found it difficult to place vessels’ root, because I think this shows that relationships between vessels and other structures in the image are being captured.

It is interesting the comment about annotator bias in the selection of the images from FGADR and it will definitely be interesting to read more about this in extensions of this work and/or at the conference.

I particularly like the segmentation experiment as downstream task, I think it would like to see these performance figures stratified per DR grade, this maybe can advise on “possibly caused by a bias in the generated data”, but shouldn’t the discriminator take care of this problem? Alternative, could embedding the downstream task to the SPADE generator end-to-end (still validating training only on real images) help SPADE produce more realistic images?  Would lesion detection + grading be a more comprehensive downstream task?


**Justification Of The Rating:**

Although there exists previous work field of generation of synthetic fundus images, I like the idea of conditioning the image generation on semantic labels. Even if there are some limitations that require further work in the extension, I think this short paper has value and deserves to be presented at MIDL.

**Paper Type:**

both

**Special Issue:**

no

---

### Official Review · Reviewer_GjHT · 2021-04-30

**Confidence:** 5
**Final Rating:** 4

**Summary:**

The authors proposed an approach to synthesize photo-realistic fundus images conditioned on synthetic “ground truth” semantic labels. The models are trained and evaluated on a combination of IDRiD and FGADR datasets. The quality of the synthetic images is validated on downstream task of lesion segmentation.

**Strengths:**

1.	The paper is well-written and easy to follow.
2.	The proposed method makes sense.
3.	I highly appreciate the downstream evaluation of such synthetic methods. The results are well presented, both in terms of quality and quantity.

**Weaknesses:**

In abstract, the authors claimed that they “demonstrate its potential for further downstream tasks, such as, but not limited to; automated grading of diabetic retinopathy, dataset balancing, creating image examples for trainee ophthalmologists, etc”. However, in reality, the only downstream task they evaluated is lesion segmentation. So, the rest of the claims should be in discussion/ future work section.

**Deanonymize Review:**

no

**Justification Of The Rating:**

This is the best short paper of my stack. Everything is logical and matured. So, MiDL will benefit from the presentation of such a well done work.
Just put the invalidated claims of abstract into the discussion/ future work section.

**Paper Type:**

both

**Special Issue:**

yes

---

### Meta-Review · Area_Chair_G259 · 2021-05-07

**Recommendation:** Accept (Poster)
**Confidence:** 5

**Metareview:**

The reviewers agree that this is a strong paper that should be published at MIDL.

---

### Decision · Program_Chairs · 2021-05-11

Accept (Poster)